# The cryo-EM structure of the human uromodulin filament core reveals a unique assembly mechanism

Jessica J Stanisich[1†], Dawid S Zyla[1†], Pavel Afanasyev[2], Jingwei Xu[1], Anne Kipp[3], Eric Olinger[4], Olivier Devuyst[3,5], Martin Pilhofer[1], Daniel Boehringer[2], Rudi Glockshuber[1]*

[1]Institute of Molecular Biology & Biophysics, ETH Zurich, Zurich, Switzerland; [2]Cryo-EM Knowledge Hub (CEMK), ETH Zurich, Zurich, Switzerland; [3]Institute of Physiology, University of Zurich, Zurich, Switzerland; [4]Translational and Clinical Research Institute, Faculty of Medical Sciences, Newcastle University, Central Parkway, Newcastle upon Tyne, United Kingdom; [5]Division of Nephrology, UCLouvain Medical School, Brussels, Belgium

*For correspondence:
rudi@mol.biol.ethz.ch

[†]These authors contributed equally to this work

Competing interests: The authors declare that no competing interests exist.

**Abstract** The glycoprotein uromodulin (UMOD) is the most abundant protein in human urine and forms filamentous homopolymers that encapsulate and aggregate uropathogens, promoting pathogen clearance by urine excretion. Despite its critical role in the innate immune response against urinary tract infections, the structural basis and mechanism of UMOD polymerization remained unknown. Here, we present the cryo-EM structure of the UMOD filament core at 3.5 Å resolution, comprised of the bipartite zona pellucida (ZP) module in a helical arrangement with a rise of ~65 Å and a twist of ~180°. The immunoglobulin-like ZPN and ZPC subdomains of each monomer are separated by a long linker that interacts with the preceding ZPC and following ZPN subdomains by β-sheet complementation. The unique filament architecture suggests an assembly mechanism in which subunit incorporation could be synchronized with proteolytic cleavage of the C-terminal pro-peptide that anchors assembly-incompetent UMOD precursors to the membrane.

The most abundant protein in human urine, the glycoprotein uromodulin (UMOD; also Tamm-Horsfall protein, THP), is conserved throughout Mammalia and produced primarily in the epithelial cells lining the thick ascending limb (TAL) of the Henle loop in the kidney. There, UMOD is important for maintaining the water impermeable layer, regulating salt transport, and urinary concentration (*Devuyst et al., 2017*). Farther along the urinary tract, UMOD has been shown to act as a soluble adhesion antagonist against uropathogenic *E. coli* (UPEC) (*Pak et al., 2001*; *Weiss et al., 2020*).

UMOD precursors traffic through the secretory pathway in the assembly incompetent pro-UMOD form, where they become glycosylated at eight potential N-glycosylation sites and eventually attached at the apical membrane surface via their glycosylphosphatidylinositol (GPI)-anchored C-terminal pro-peptide (CTP) (*Rindler et al., 1990*; *van Rooijen et al., 1999*; *Weiss et al., 2020*). The protease hepsin then cleaves the CTP and UMOD assembles to homopolymeric filaments with an average length of 2.5 µm in the urine (*Brunati et al., 2015*; *Porter and Tamm, 1955*).

Recently, the general architecture of the intrinsically flexible UMOD filaments was uncovered via cryo-electron tomography (*Weiss et al., 2020*). The results, together with previous data, showed that UMOD assembles to a zigzag shaped, linear polymer, where the core of the filament is formed by its bipartite zona pellucida (ZP) module with the subdomains ZPN and ZPC. The N-terminal domains, epidermal growth factor-like (EGF) I–III and the following cysteine-rich D8C domain, protrude as arms alternating from opposite sides of the filament, and the EGF IV domain connects the arms to the filament core (*Figure 1A*; *Jovine et al., 2002*; *Schaeffer et al., 2009*; *Weiss et al.,*

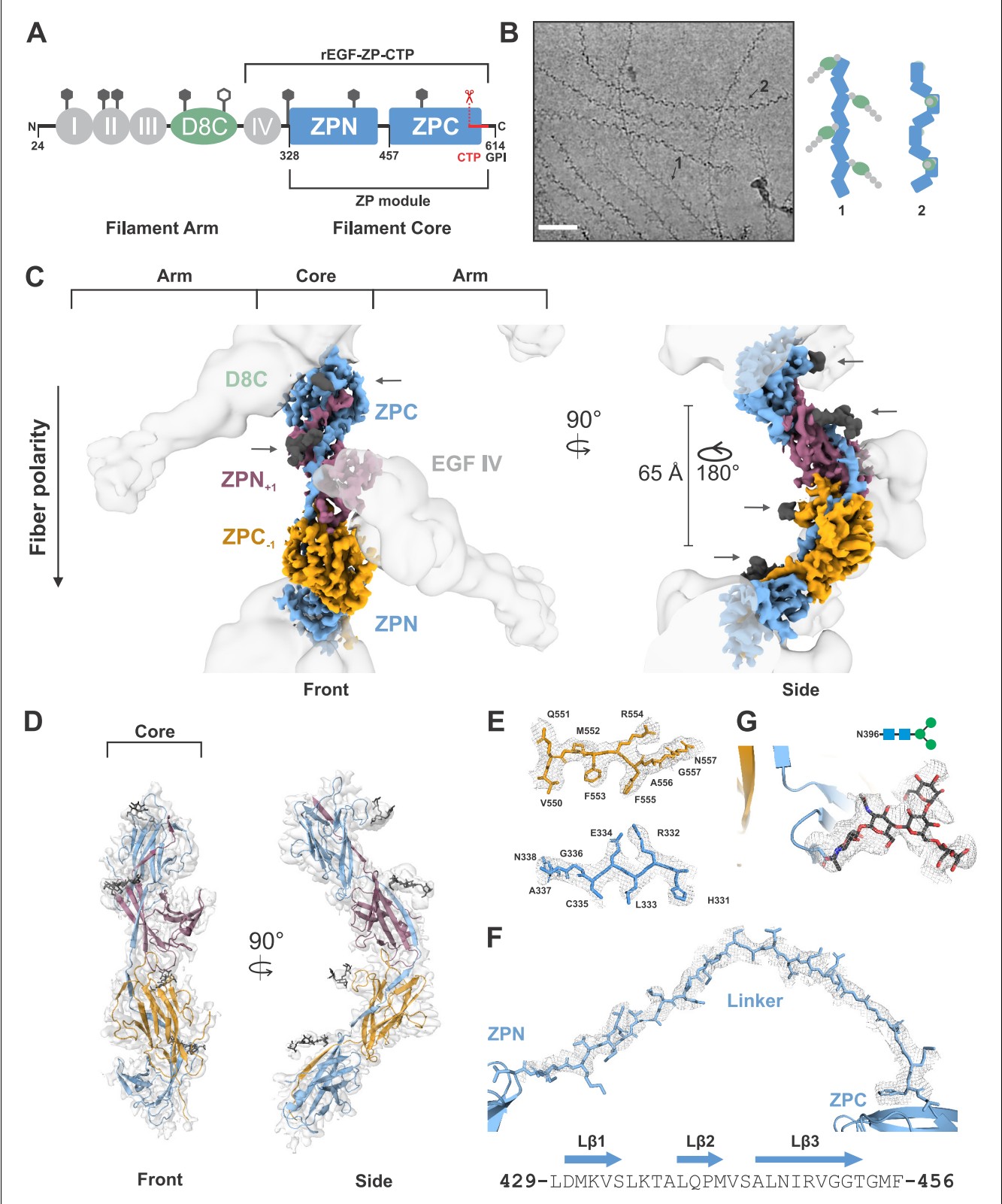

**Figure 1.** Cryo-EM structure of the human UMOD filament core. (**A**) Domain architecture of the membrane-anchored pro-UMOD monomer, composed of 4 EGF-like domains (I-IV, gray), the cysteine-rich D8C domain (green), and the bipartite ZP module (ZPN and ZPC, blue). The fold of the C-terminal ZPC subdomain is extended by the GPI-anchored, C-terminal pro-peptide (CTP, red) that is cleaved by hepsin as a prerequisite of UMOD polymerization. The hepsin cleavage site (peptide bond 587–588) is indicated by scissors. The previously crystallized UMOD segment rEGF-ZP-CTP is

*Figure 1 continued on next page*

*Figure 1 continued*

also indicated. The positions of the eight verified N-glycosylation sites in mature UMOD filaments are indicated above the respective domains as hexagons (filled hexagons: complex type N-glycan; open hexagon: high mannose type N-glycan). Amino acid numbering according to pre-pro-UMOD including signal sequence. (B) Representative cryo-electron micrograph showing the two major views of UMOD filaments: front view (1) and side view (2). Scale bar: 50 nm. Schematic representations of the subdomain organization of these two views are shown on the right. (C) Segmented Coulomb potential map of the mature ZP module in the filament, shown over the low-pass filtered reconstruction (at ~9 Å resolution, gray). The extended linker connecting ZPN and ZPC of the blue ZP module complements the folds of ZPN of the following and ZPC of the preceding ZP modules, ($ZPN_{+1}$ and $ZPC_{-1}$; plum and gold, respectively) by β-sheet complementation. The map features show a helical rise of 65 Å and a twist of 180°. The locations of the EGF IV and D8C domains from the filament arms are also indicated. The resolved N-glycan map features (dark grey) are indicated with arrows. (D) Front and side views of the refined model of the filament core within the obtained high-resolution cryo-EM map. Resolved monosaccharide units of N-glycans are shown as gray stick models. (E) Detailed view of the map features at selected regions in ZPN (blue) and ZPC (gold), illustrating the quality of the final cryo-EM model. (F) Contiguous Coulomb potential map (mesh) around the extended linker segment that harbors the β-strands $L\beta1$–$L\beta_3$. $L\beta_1$ and $L\beta_2$ complement the ZPC fold of the preceding UMOD subunit, and $L\beta_3$ complements ZPN of the following UMOD subunit. (G) Coulomb potential map around the core pentasaccharide of the complex type N-glycan attached to N396 in ZPN. Blue squares: N-acetylglucosamine; green circles: mannose. The online version of this article includes the following video and figure supplement(s) for figure 1:

**Figure supplement 1.** Cryo-EM data evaluation for UMOD filaments.
**Figure supplement 2.** Representative image of the high-resolution cryoSPARC Coulomb potential map fitted with the asymmetric unit (AU) UMOD model.
**Figure supplement 3.** Flexibility analysis of native UMOD filaments.
**Figure 1—video 1.** Cryo-EM map of a native UMOD filament.
https://elifesciences.org/articles/60265#fig1video1
**Figure 1—video 2.** 3D variability analysis of UMOD filaments.
https://elifesciences.org/articles/60265#fig1video2

*2020*). UMOD filaments encapsulate piliated uropathogens through multivalent binding via their N-glycans to the lectins at the tips of bacterial pili. This mechanism not only prevents pathogen adhesion to target glycans on epithelial cells of the host, but also facilitates pathogen aggregation and clearance by urine excretion (*Weiss et al., 2020*).

Despite these insights, the exact structural organization of UMOD filaments remained unknown. We set out to solve the structure of the UMOD filament by single-particle cryo-EM. Isolated native UMOD filaments from a healthy individual appeared on the grid in two major views: a fishbone-like 'front' view with alternately protruding arms, and a zigzag 'side' view (*Figure 1B*). Marked conformational variation in both the filament core and the angles between the peripheral arms and the core was visible. Additionally, flexible N-glycans lining the filament core could be seen already in 2D class averages (*Figure 1—figure supplement 1*). This intrinsic flexibility and glycan display is a prerequisite of pathogen encapsulation by UMOD filaments, however, they considerably complicated the structure determination by single-particle cryo-EM, restricting high-resolution analysis to the filament core.

*Ab initio* 3D reconstruction of 723,000 particles was followed by extensive classification and focused refinement together with the non-uniform refinement strategy in cryoSPARC in order to obtain a reconstruction of the UMOD core at 3.5 Å (*Figure 1—video 1*). This high-resolution, unsymmetrized map was used to rigid body fit the individual UMOD ZP subdomains, ZPN and ZPC, from the previously published crystal structure of a recombinant, assembly incompetent pro-UMOD fragment (consisting of the EGF IV domain, ZP module, and CTP (termed rEGF-ZP-CTP; *Bokhove et al., 2016*; PDB: 4WRN). The resolution of the cryoSPARC map at the filament core allowed direct model building of the immunoglobulin-like (IG) ZP subdomain folds. The ZPN/ZPC asymmetric unit was then extended by rigid-body fitting within a lower resolution map (4.7 Å), refined in cisTEM, that contained more repeats. This fitting showed a helical architecture of the filament with a ~ 65 Å rise and ~180° twist, which is consistent with parameters obtained by sub-tomogram averaging (*Weiss et al., 2020*). From here, we were able to ascertain an unexpected, extended conformation of the ZP monomers in the filament core, wherein a long, 28-residue linker (UMOD residues L429–F456) between the ZPN and ZPC subdomains of each ZP monomer spanned the ZPC from the previous module ($ZPC_{-1}$) and the ZPN from the subsequent module ($ZPN_{+1}$) (*Figure 1C–D*). The core region including the linker is well resolved, showing clear sidechain map features (*Figure 1E–F*). The linker between ZPN and ZPC of the monomers spans ~103 Å and interacts tightly with the two intervening ZP subdomains (*Figure 1D–F*, *Figure 1—figure supplement 2*). Moreover,

Coulomb potentials for prominent N-glycans at N396 and N513 were observed and allowed visualization of the core pentasaccharide and disaccharide, respectively (*Figure 1C,D,G*).

The intrinsic flexibility of the UMOD fibers could be directly observed in the collected micrographs in the range of their curvatures (*Figure 1B*). Slight deviations in the helical rise along the chain and the intrinsic flexibility hampered particle classification and alignment and limited the resolution of the reconstructions. To analyze this flexibility, we measured the relative angles between subdomains after rigid-body docking into multiple, heterogeneous 3D class averages (*Figure 1—figure supplement 3*). Similar maximum angular changes of ~7 degrees were obtained at both unique interfaces between the ZP subdomains, indicating that both interfaces equally give rise to the flexibility of the filaments. Additionally, we utilized the 3D variability analysis from cryoSPARC (based on the principle component analysis (PCA); *Punjani and Fleet, 2020*), which showed a similar degree of filament bending (component 1) and an elongation of the filament along the z-axis (component 2) based on a relative rotation of ZP modules at the subdomain interfaces (*Figure 1—video 2*).

The elongated ZP linker establishes an intricate and extensively enchained scaffold for filament build-up from mature UMOD monomers. Reminiscent of donor strand complementation in subunit-subunit interactions of pili assembled via the chaperone-usher pathway (*Waksman, 2017*) or the more recently discovered type V pili (*Shibata et al., 2020*), formation of the UMOD filament involves a β-sheet complementation mechanism extending across enchained ZP modules (*Figure 2A*). Specifically, the extended linker between ZPN and ZPC in each monomer is comprised of three separate β-strands: $L\beta_1$, $L\beta_2$, and $L\beta_3$ (residues 430–435, 438–441, and 446–452, respectively), where $L\beta_1$ and $L\beta_2$ of subunit n complement the fold of ZPC of subunit n–1 ($ZPC_{-1}$), and $L\beta_3$ extends the IG-fold of ZPN of subunit n+1 ($ZPN_{+1}$) (*Figure 2B*). The complementation site (CS) between $L\beta_1$, $L\beta_2$, and $ZPC_{-1}$ (denoted $CS_A$) is formed by the antiparallel insertion of $L\beta_1$ relative to the ZPC $\beta_F$ strand, and the shorter, parallel complementation of the ZPC $\beta_{A'}$ strand by the $L\beta_2$ strand (*Figure 2C*). $L\beta_3$, the longest β-strand in the linker segment, binds parallel along the $ZPN_{+1}$ $\beta_G$ strand at complementation site B ($CS_B$) (*Figure 2D*).

The inter-molecular interface between the ZPN subdomain of subunit n and the ZPC domain of the preceding subunit ($ZPC_{-1}$) (interface A, $I_A$), is formed by mostly hydrophobic residues from both subunits (*Figure 2E*), creating a buried surface area of ~2000 Å$^2$ (PISA server, *Krissinel and Henrick, 2007*). Specifically, residues Y402, Y427, and L429 from ZPN at $I_A$ cover a prominent hydrophobic patch on the surface of the $ZPC_{-1}$, composed of L491, F499, F456, and L570 (*Figure 2—figure supplement 1*). The second unique intermolecular interface ($I_B$) along the elongated UMOD monomer is formed between $ZPC_{-1}$ and $ZPN_{+1}$, and is spanned by the linker (*Figure 2F*). Surface charge complementary between the two ZP subdomains provides the basis for $I_B$, highlighted by the insertion of R415 on $ZPN_{+1}$ into a negatively charged pocket on $ZPC_{-1}$ harboring D532. $I_B$ is further stabilized by hydrophobic interactions between $ZPC_{-1}$ (F555, F553) $ZPN_{+1}$ (I413, I414) and the linker (P441, V443) (*Figure 2—figure supplement 1*).

We utilized the Genome Aggregation Database (gnomAD) to parse missense variants in the UMOD ZP module detected in ~125,000 exome and ~15,000 whole-genome sequences from unrelated healthy individuals from various population studies (*Karczewski et al., 2020*). Given the selective pressure for high levels of functional UMOD filaments in the urine (*Ghirotto et al., 2016*), we reasoned that residues or regions of UMOD that are required for filament assembly should show less genetic variation in healthy individuals. Here, we found that both the linker segment and those residues important for creating $I_A$ and $I_B$ are depleted in genetic variants, realtive to the rest of the ZP module (*Figure 2—figure supplement 2*; *Supplementary file 1A*).

The previously reported structure of the pro-UMOD fragment rEGF-ZP-CTP, which crystallized as a homodimer in which only the ZPN subdomains interact (*Bokhove et al., 2016*), provides the nearest comparison for the transition from pro-UMOD (CTP intact) to mature UMOD in the filament. *Figure 3* shows the major conformational differences between rEGF-ZP-CTP and the ZP module in the mature filament. The most prominent difference is the separation of ZPN and ZPC in the mature filament, where individual C$^\alpha$ atoms (e.g. ZPN residue S364) change their relative position by up to 95 Å (*Figure 3A*). In addition, formation of the 103 Å long linker segment in mature UMOD involves i) the excision of strand $\beta_A$ from ZPC, which becomes the linker strand $L\beta_3$ in the mature ZP module, and ii) the unwinding and extension of the α-helical segment connecting ZPN and ZPC in pro-UMOD that forms linker strands $L\beta_1$ and $L\beta_2$ in the mature ZP module (*Figure 3B*; *Figure 3—figure supplement 1*). The overall folds of the ZPN and ZPC subdomains, however, remain the same (with

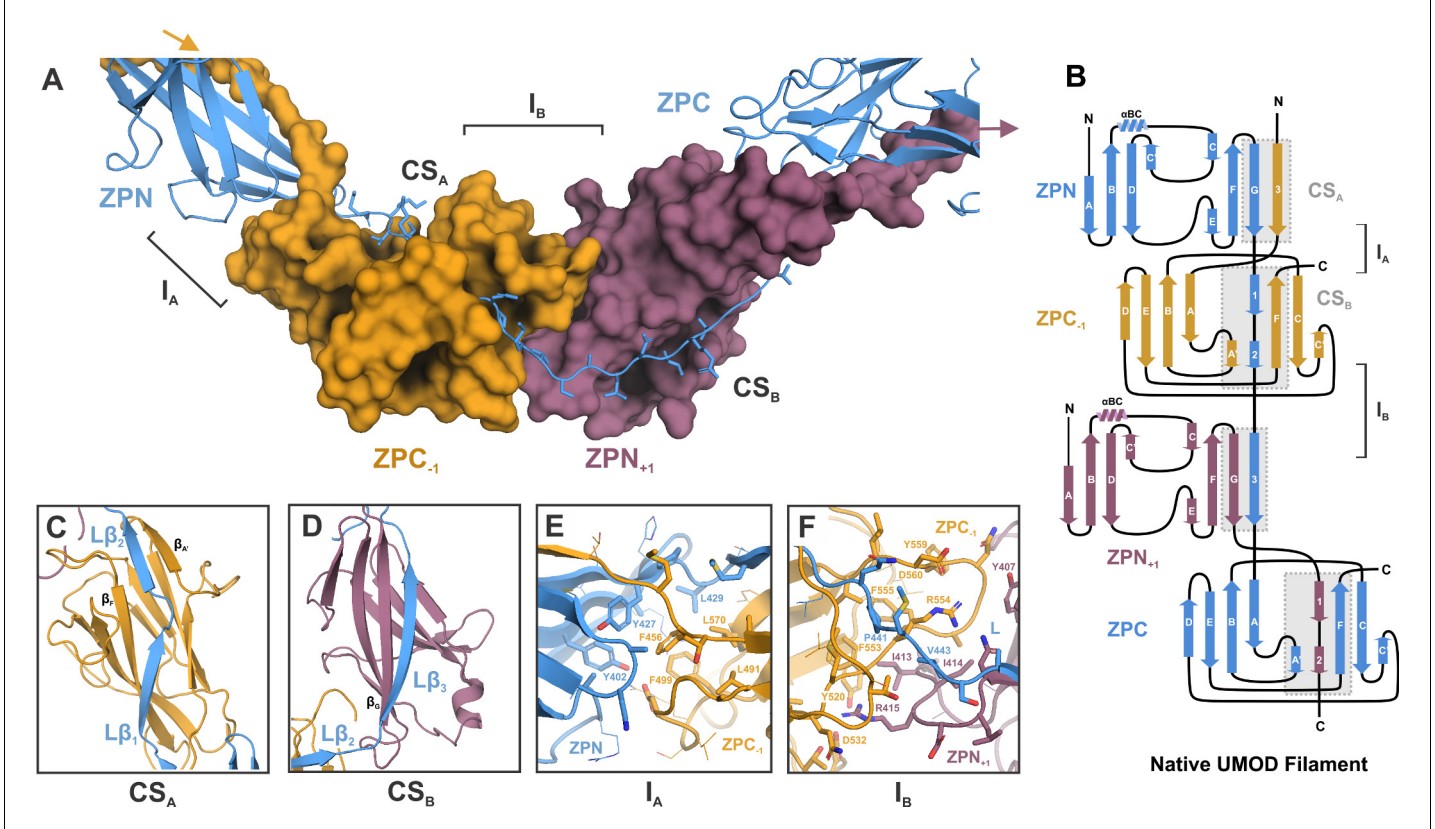

**Figure 2.** Inter-molecular β-sheet complementation in the ZP subdomains results in extensive interactions between neighboring subunits in the UMOD filament. (A) Cartoon representation of the ZP module of subunit n (blue) in complex with subdomain $ZPN_{+1}$ (plum) and the subdomain $ZPC_{-1}$ (gold) as surface representations. The total length of the polypeptide backbone of the extended 28-residue linker between ZPN and ZPC (stick representation) is approximately 103 Å. The β-sheet complementation sites $CS_A$, between the linker strands $Lβ_1/Lβ_2$ and $ZPC_{-1}$, and $CS_B$, between linker strand $Lβ_3$ and $ZPN_{+1}$, as well as the interfaces between the subdomains ZPN and $ZPC_{-1}$ (interface $I_A$) and $ZPN_{+1}$ and ZPC (interface $I_B$) are indicated. (B) Topology model indicating secondary structure elements of mature UMOD filaments based on the solved cryo-EM structure. (C) Complementation site A ($CS_A$): the linker strands $Lβ_1$ and $Lβ_2$ complement the incomplete fold of $ZPC_{-1}$ by interacting with its strand $β_F$ and $β_{A'}$, respectively. (D) Complementation site B ($CS_B$): the linker strand $Lβ_3$ complementing the fold of $ZPN_{+1}$ by interaction with its strand $β_G$. (E) Interface A ($I_A$) between $ZPN_n$ (blue) and $ZPC_{-1}$ (gold) showing extensive intermolecular hydrophobic interactions between the subdomains. Residues forming the interface are shown as sticks and labeled with the corresponding residue numbers (same color code as in A). (F) Interface B ($I_B$) between $ZPC_{-1}$ (gold), $ZPN_{+1}$ (plum) and the linker peptide (blue) is stabilized by electrostatic and hydrophobic interactions. Loops of the ZP subdomains meet as a coupler with the linker spanning across.

The online version of this article includes the following figure supplement(s) for figure 2:

**Figure supplement 1.** UMOD ZP module interfaces are stabilized by hydrophobic packing.

**Figure supplement 2.** gnomAD zero variance residues of the mature UMOD fiber linker region.

$C^α$ RMSD values of 2.0 and 1.2 Å, respectively). Notably, linker strand $Lβ_2$ occupies the same position in $ZPC_{-1}$ as the β-strand segment (559–605) from the CTP in rEGF-ZP-CTP. In the mature filament, the residues that constitute helix αEF of rEGF-ZP-CTP are extended and the involved aromatic residues undergo a rearrangement that allows binding of $Lβ_2$ (*Figure 3—figure supplement 2*). In addition, the insertion of the linker strand $Lβ_1$ into $ZPC_{-1}$ only becomes possible after the excision of its strand $β_A$, which blocks $Lβ_1$ binding in rEGF-ZP-CTP (*Figure 3C–E*). Overall, the binding pocket for the CTP β-strand in ZPC of rEGF-ZP-CTP is shorter than that accommodating $Lβ_1$ and $Lβ_2$ in the mature ZPC subdomain, causing a kink in the CTP peptide along the ZPC subdomain (*Figure 3E*). The difference observed in comparison of the ZPN domains is less dramatic; here, the same surface complemented by the rEGF-ZP-CTP strand $β_G$ from the apposing ZPN in the crystal structure homodimer is fulfilled in the filament by the $Lβ_3$ strand from subunit n–1 (*Figure 3F,G*).

The cryo-EM structure of the UMOD filament core raises the intriguing question of how UMOD filaments assemble from pro-UMOD subunits attached to the cell surface. *Figure 4* shows the

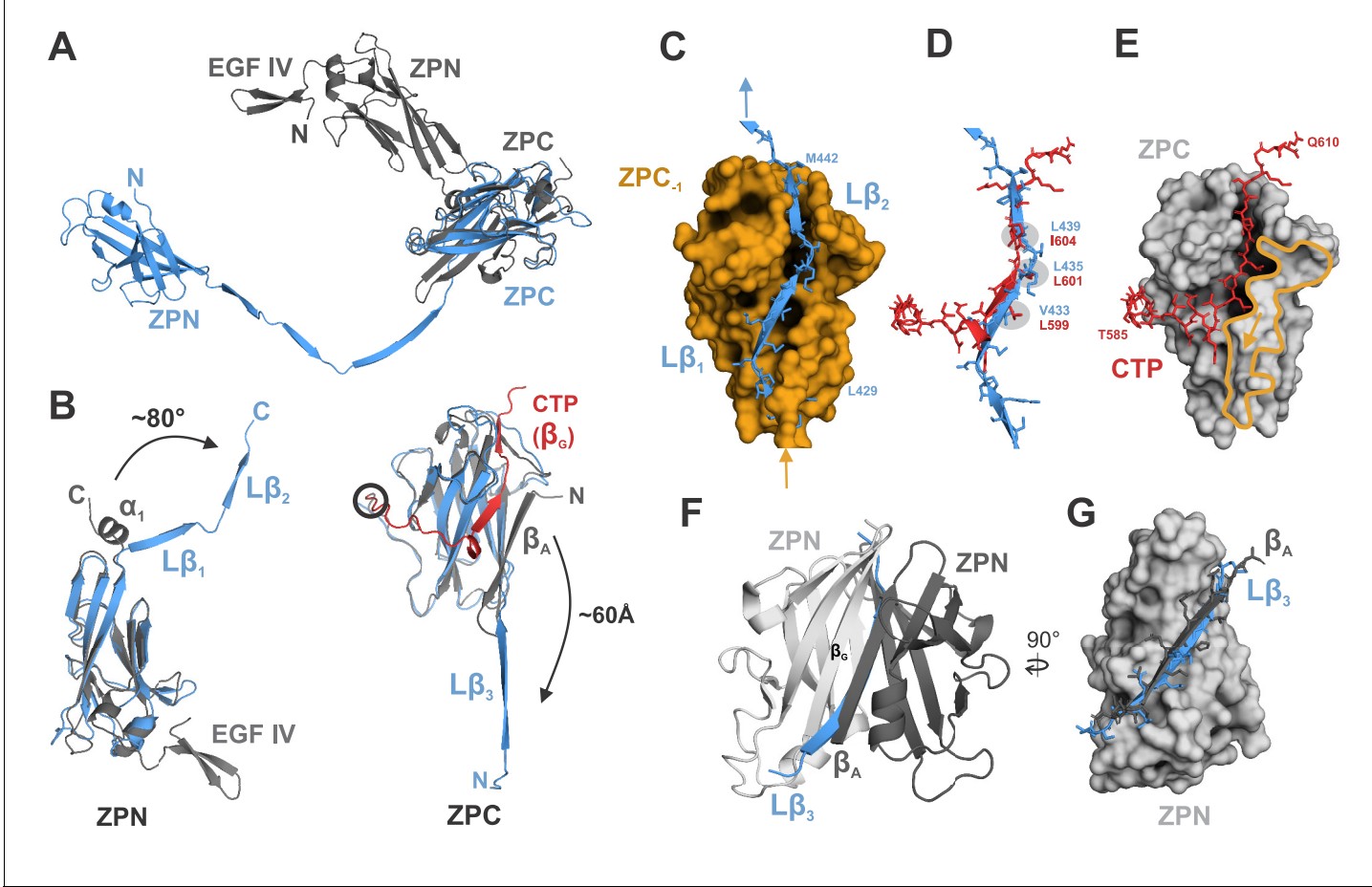

**Figure 3.** Comparison of ZP module structure in the native filament to that in the crystal structure of rEGF-ZP-CTP. (**A**) Superposition of the structure of the ZP module in the rEGF-ZP-CTP homodimer (gray, PDB: 4WRN) with the solved cryo-EM structure of the mature ZP module in the filament (blue). Both structures were aligned via the ZPC subdomains. In the filament, the ZPN subdomain is shifted away from the ZPC subdomain of the same subunit by ~60 Å. (**B**) Rearrangement of secondary structure elements in ZPN and ZPC forming the extended linker upon UMOD polymerization compared to rEGF-ZP-CTP. The cryo-EM model of mature UMOD core is shown in blue, and the X-ray structure of rEGF-ZP-CTP (PDB: 4WRN) in gray. The CTP (absent in the mature filament) is highlighted in red. Left: The α-helical segment at the C-terminus of ZPN in pro-UMOD forms the linker strands $L\beta_1$ and $L\beta_2$ in the mature filament. Right: The N-terminal ZPC β-strand ($\beta_A$) in pro-UMOD forms the strand $L\beta_3$ in the extended linker of the mature filament. The CTP (red), providing the last ZPC β-strand ($\beta_G$) in pro-UMOD. The hepsin cleavage site in pro-UMOD is circled. (**C**) Surface representation of the ZPC subdomain (gold) with the two complementing strands $L\beta_1$ and $L\beta_2$ from the following subunit. Labeled residues indicate the beginning and the end of $L\beta_{1-2}$. (**D**) Superposition of the C-terminal pro-peptide forming pro-UMOD ZPC strand $\beta_G$ (red) with the linker strands $L\beta_1$ and $L\beta_2$ (blue) that occupy the same hydrophobic groove in the polymer ZPC. Similar or identical hydrophobic β-strand sidechains are bound to the hydrophobic sidechain pockets in ZPC (grey) in both structures. (**E**) Binding groove of pro-UMOD CTP (red, sticks) on the ZPC (gray surface). The N-terminal ZPC strand $\beta_A$ (yellow outline) limits the size of the binding pocket. The $\beta_A$ strand is flipped out upon conformational change from pro-UMOD mature UMOD (then $L\beta_3$). (**F**) Structure of the crystallized ZPN homodimer of pro-UMOD (PDB: 4WRN). The N-terminal strand $\beta_A$ of each ZPN subdomain interacts with the C-terminal $\beta_G$ strand of the opposite ZPN. (**G**) In the UMOD polymer, ZPN $\beta_G$ interacts with linker strand $L\beta_3$ (blue) instead, which occupies the same position as strand $\beta_A$ (black) from the apposing subunit in the dimer.

The online version of this article includes the following figure supplement(s) for figure 3:

**Figure supplement 1.** Regular secondary structure topology diagrams for the ZP modules.

**Figure supplement 2.** Rearrangement of aromatic residues at interface $I_B$.

simplest conceivable model of UMOD filament assembly. Assembly requires the release of mature UMOD monomers from the membrane by proteolytic removal of the GPI-anchored CTP by hepsin, thus making a ZPC module competent for binding the $L\beta_3$ strand of another UMOD. Assembly starts by binding of a pro-UMOD ZPN module to strands $L\beta_1$ and $L\beta_2$ in a neighboring pro-UMOD (step 1). Hepsin cleavage may then occur at the elongating filament (step 2). Each subsequent UMOD incorporation step (steps 3, 4, etc.) would then be characterized by the intercalation of the ZPN

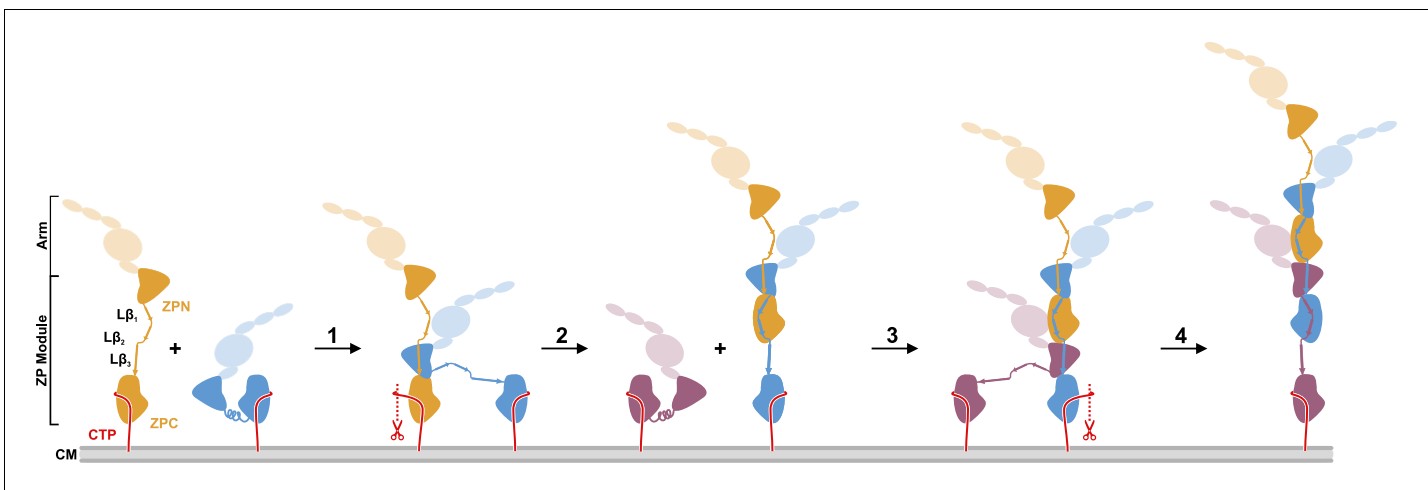

**Figure 4.** Proposed model of UMOD polymerization after hepsin cleavage of pro-UMOD. UMOD arms with the domains EGF I-III, D8C, and EGF IV are shown transparent to indicate their positions relative to the ZP domains. (Step 1) Binding of ZPN from a pro-UMOD monomer (blue) to the L$\beta_3$ segment of an extended neighbor pro-UMOD (gold) may start the assembly. (Step 2) In the resulting, asymmetric pro-UMOD dimer, hepsin (red scissors) cleaves the GPI anchored CTP (red) from the ZPC subdomain. The released ZPC subdomain then binds to L$\beta_1$ and L$\beta_2$ of the incoming pro-UMOD. (Step 3) ZPN from a third pro-UMOD (plum) binds to the L$\beta_3$ segment between the two ZPC segments of the growing filament. (Step 4) Again, hepsin cleaves off the CTP from the pro-UMOD in which the ZPC is complexed with ZPN. Steps 3 and 4 are consistently repeated until filament assembly is completed.

The online version of this article includes the following figure supplement(s) for figure 4:

**Figure supplement 1.** MALDI-MS/MS analysis of tryptic peptides of mature human UMOD filaments.

**Figure supplement 2.** Amino acid sequence alignment of ZP-containing, GPI anchored human proteins.

from an incoming pro-UMOD between the two ZPC modules of the filament, leading to a more stable binding of the incoming ZPN compared to that in step 1. As a final cleavage step to release the assembled filaments from the membrane, a yet unidentified hydrolase may act upon UMOD, as small amounts of the CTP could still be detected in MS/MS spectra after trypsin digestion of mature filaments (*Figure 4—figure supplement 1*). This UMOD assembly model is analogous to the recently proposed assembly mechanism of filamentous type V pili, in which assembly is linked with proteolytic release of pilus subunits from the outer bacterial membrane (*Shibata et al., 2020*). During the preparation of this manuscript, a related preprint article on the cryo-EM structure of the UMOD filament core was published (*Stsiapanava et al., 2020*). In said study, an alternative model of UMOD assembly was proposed, based on the assumption that assembly starts from membrane-bound pro-UMOD homodimers.

The enchained β-sheet complementation mechanism observed in the UMOD filament core may also be valid for other proteins containing membrane-anchored ZP modules and undergoing filament formation. For instance, another human ZP protein, alpha-tectorin (TECTA), forms filaments creating the basis for the apical extracellular matrix (ECM) on the cochlear supporting cells and shares important features with UMOD: a highly similar linker region and a conserved, C-terminal protease cleavage site (*Figure 4—figure supplement 2*). Recently, Kim and colleagues proposed the '3D printing model' for surface-tethered, TECTA-mediated ECM organization (*Kim et al., 2019*), which is in agreement with our proposed model of UMOD polymerization at the surface of TAL cells in the kidney tubule. Thus, the cryo-EM structure of the UMOD filament core might be representative for the core structure of multiple proteins with a C-terminal ZP module that become functional after polymerization.

## Materials and methods

### Purification of human UMOD

Human UMOD fibers were purified from a healthy donor as described in *Weiss et al., 2020*. Briefly, urine was purified using a diatomaceous earth filter, concentrated, and dialyzed overnight against 0.5 mM EDTA-NaOH, pH 8.2 (300 kDa cut-off, Spectrum laboratories). Aliquots were flash-frozen in liquid nitrogen and stored at –20°C until further use. Only the second morning micturition was utilized for urine collection. Protein concentrations were determined as previously described (*Weiss et al., 2020*).

### Cryo-EM data collection

Protein samples (3.5 µL of 2.5 µM UMOD) were applied to glow-discharged lacey carbon grids with an ultrathin carbon coating (Electron Microscopy Sciences) automatically blotted from the back side of the grid for 13.5 s (100% humidity, 9°C) and plunge frozen in liquid-ethane-propane using a Vitrobot Mark IV (Thermo Fisher Scientific). Micrographs were acquired on a Titan Krios microscope (Thermo Fisher Scientific) operated at 300 kV with a Gatan K2 Summit direct electron detector using a slit width of 20 eV on a GIF-Quantum energy filter. A total of 4679 and 4864 movies were recorded over two data collections from two separate grids and subsequently merged. Images were recorded with EPU software (Thermo Fisher Scientific) at 130 000 x nominal magnification in counting mode with a calibrated pixel size of 1.084 Å. The defocus target ranges were −1.2 µm to −3.3 µm and −0.8 µm to −2.0 µm for the first and second data collection, respectively. Each micrograph was dose-fractionated to 40 frames under a dose rate of 7.5 e$^-$ / Å / s, with an exposure time of 6 s, resulting in a total dose of approximately 45 e$^-$ / Å$^2$.

### Cryo-EM image processing

The collected movies were motion corrected with MotionCor2 (*Zheng et al., 2017*). The CTF parameters of the micrographs were estimated using Gctf 1.06 (*Zhang, 2016*). Following steps of image processing were done in cryoSPARC v2.15 (*Punjani et al., 2017*) and with the in-house written python scripts for data administration and interpretation (https://github.com/dzyla/umod_processing_scripts). Approximately 600 particles were manually picked from a random selection of micrographs and used to train a convolutional neural network picking model for TOPAZ (*Bepler et al., 2019*). A total of 1.3 million particles were picked via TOPAZ from all micrographs. For the first steps of processing, particles were extracted with 320 px box size and binned to 2.71 Å/px. After several rounds of 2D classification, 723,000 particles, corresponding to well resolved, clear classes, were selected for further processing. Three *ab initio* initial models were generated and a single best one was used as a 3D-reference for the 3D-heterogeneous refinement. 3D-heterogeneous refinement with 10 classes resulted in multiple distinct 3D class-averages, showing intrinsic flexibility of the specimen. The most populated 3D class-average of 145,000 particles was chosen for further refinements. Particles from the selected class were then re-extracted at the original pixel size and subjected to a 3D-homogenous refinement. A mask was created in UCSF Chimera (*Pettersen et al., 2004*) for the center segments of 3D-reconstruction and used for several rounds of local refinement using a non-uniform refinement strategy. This pipeline yielded high-resolution 3D-reconstruction of the central asymmetric unit (AU; ZPN, ZPC, and the inset linker) at 3.5 Å. Detailed workflows are shown in *Figure 1—figure supplement 1*, *Supplementary file 1B,C*.

To obtain a map focused on several repeat units, an alternative approach was applied using cisTEM for refinement and classification: all micrographs were manually selected, resulting in a total of 8543 movies which were then also motion corrected using MotionCor2, CFT values estimated using Gctf, and 485,000 particles were manually picked as helices from a random selection of 3600 micrographs in RELION v3.0.8 (*Zivanov et al., 2018*) and extracted at the original pixel size. The particle stack was then subjected to 2D-classification in cisTEM (*Grant et al., 2018*). 402,000 good particles were selected, and multiple rounds of focused classification was performed with the imported best cryoSPARC 3D class-average as a reference (described above) and a spherical (120 Å radius) mask. Two of the resulting 3D class-averages were merged and another round of 3D-classification with the same spherical mask was undertaken, resulting in a map resolved to 4.7 Å from 330,000 particles. The map was filtered to 4.7 Å and sharpened using cisTEM default settings (pre-cut-off B-Factor

−90.0, resolution cut-off 4.7 Å). Reported helical parameters were estimated in real space from the cisTEM map using *relion_helix_toolbox* (*He and Scheres, 2017*) with search ranges for rise (60–67 Å) and twist (175–185°). Schematic workflows and details are shown in *Figure 1—figure supplement 1*, *Supplementary file 1B,D*. For comparative purposes, the cisTEM map FSC curves and the local resolution maps were calculated in cryoSPARC for all generated half-maps.

## Model building and structure refinement

The individual ZP subdomains of the UMOD crystal structure (PDB: 4WRN; *Bokhove et al., 2016*) were used as initial models (ZPN: aa 428–528, ZPC: aa 541–710, according to the 4WRN numbering). First, the individual domains were rigid-body fitted to the map obtained in cryoSPARC using UCSF Chimera, followed by manual building in Coot 0.9 and ISOLDE (*Croll, 2018*; *Emsley et al., 2010*). The asymmetric unit (AU, defined as a single ZPN and ZPC and the inset linker) of UMOD was refined using *phenix.real_space_refine* program (*Adams et al., 2010*) with the default settings, which apply Ramachandran restraints. To reduce clashes between subunit interfaces, the AU model was used to create a model with four consecutive ZP subdomains and was refined against the cisTEM map. The structure of the UMOD ZP module was then applied to the RosettaCM refinement with the symmetry derived from the cisTEM map (*Song et al., 2013*), with the 'auto' strategy applied in the refinement. The central ZPN/ZPC module in the refined structure was manually adjusted to the high-resolution cryoSPARC map in COOT and was subsequently used to generate the final UMOD model. This updated model was subjected to the next round of RosettaCM refinement, with a higher RMS model restraint applied (r.m.s. = 0.2). Overall, three rounds of interactive manual and RosettaCM refinements were performed and the geometry parameter of the final AU model structure was further improved using *phenix.real_space_refine* with tight reference restraints (sigma = 0.025). The sidechain residues 462–473 in the AU model could not be built with confidence and were omitted from the final model.

   For the final model of the UMOD filament, a structure was generated with 3 copies of the final AU model by applying the symmetry derived from the cisTEM map. The central ZPN/ZPC subunit of the core part of the structures were subjected to a real space refinement with tight reference restraints (sigma = 0.025) using the cryoSPARC map. The center ZPN/ZPC subunit in the refined model was then used to generate the new UMOD filament structure. Clashes in the subunit interface were further reduced by performing another round of *phenix.real_space_refine* with tighter reference restraints (sigma = 0.02).

   Model validations were performed using the Comprehensive Validation tool in PHENIX. Final model statistics were generated using MolProbity tool via PHENIX (*Chen et al., 2010*). Figures of the structure were prepared with PyMol v2.4 (The PyMOL Molecular Graphics System, Version 2.4 Schrödinger, LLC) and EM map figures were generated with UCSF ChimeraX 0.93 (*Goddard et al., 2018*).

## Cryo-EM model 3D flexibility analysis

The well-resolved reconstructions from 3D classifications steps performed independently in cryoSPARC and cisTEM were rigid-body fitted with ZP subdomains derived from the final model. Each subdomain was first placed at the correct position and then fitted by the *Fit in Map* tool in Chimera 1.13.1. In total, 6 ZP subdomains were fitted into nine maps where the subdomain Coulomb potential was well defined (four classes from cryoSPARC and five classes from cisTEM) and saved with fixed positions to each other as a 'filament'. Next, using the PyMOL *align* command, all nine filament structures were aligned to the center ZPN subdomain. In Chimera, the *define axis* command was used to give each subunit a centroid and a central axis. For subunits at positions $ZPC_{-1}$ and ZPC, the maximum angle was determined with the *angle* command.

## 3D variability analysis of UMOD

3D variability analysis (3DVA) was performed in cryoSPARC using ~723,000 particles extracted at the pixel size of 4.34 Å, 10 modes, and filter resolution of 9.2 Å. The 3DVA display job was run in the simple output mode with 20 intermediate frames. From 10 generated components, the first two exhibited the same yaw/pitch movement (*Figure 1—video 2*), the third showed an artifact at the

box edge, and the fourth showed the elongation along the filament axis (*Figure 1—video 2*). The higher components showed various movements in the arms (not shown).

### Assessing genetic variation using gnomAD

Genome Aggregation Database (GnomAD) v2.1.1 (https://gnomad.broadinstitute.org/) is aligned against the GRCh37 genome build and was released in March 2019. The dataset comprises 125,748 exomes and 15,708 genomes sequenced as part of various disease-specific and population genetic studies, totaling 141,456 unrelated individuals from eight major populations (*Karczewski et al., 2020*). Results for *UMOD* were filtered for missense variants and checked for consistency with the *UMOD* transcript ENST00000302509.8. Missense variants corresponding to a non-canonical transcript were manually removed.

UMOD variants were parsed using an in-house Python script (https://github.com/dzyla/umod_processing_scripts) and plotted on the structure. Residues with no mutations were set to 1.0 and for all non-zero variants the values were normalized between 0 and 1, based on the PAM250 matrix. The structure was visualized by spectrum coloring in PyMOL 2.4. For *Supplementary file 1A*, all linker residues are shown and the interfaces are defined by atoms that are within 8 Å of one another, as defined by COCOMAPS (*Vangone et al., 2011*).

## Acknowledgements

The authors thank M Peterek for technical support during EM data collection and the ScopeM platform for instrument access at ETH Zurich. We also thank the Functional Genomics Center Zurich, specifically P Hunziker, for the MS/MS analysis. We would like to acknowledge R Zdanowicz for early support screening cryo-EM grids. RG was supported by the Swiss National Science Foundation (310030B_176403/1 and 31003A_156304) and basic funding by ETH Zurich. MP was supported by the Swiss National Science Foundation (31003A_179255), the European Research Council (679209) and the NOMIS foundation. OD and AK are supported by the Swiss National Centre of Competence in Research Kidney Control of Homeostasis (NCCR Kidney.CH) program and the Swiss National Science Foundation (310030_189044). EO is supported by an Early Postdoc Mobility-Stipendium of the Swiss National Science Foundation (P2ZHP3_195181).

## Additional information

### Funding

| Funder | Grant reference number | Author |
|---|---|---|
| Schweizerischer Nationalfonds zur Förderung der Wissenschaftlichen Forschung | 310030B_176403/1 | Rudi Glockshuber |
| Schweizerischer Nationalfonds zur Förderung der Wissenschaftlichen Forschung | 31003A_156304 | Rudi Glockshuber |
| Schweizerischer Nationalfonds zur Förderung der Wissenschaftlichen Forschung | 31003A_179255 | Martin Pilhofer |
| H2020 European Research Council | 679209 | Martin Pilhofer |
| NOMIS Stiftung | | Martin Pilhofer |
| Schweizerischer Nationalfonds zur Förderung der Wissenschaftlichen Forschung | 310030_189044 | Olivier Devuyst |
| Swiss National Centre of Competence in Research Kidney Control of Homeostasis | | Olivier Devuyst |
| Schweizerischer Nationalfonds zur Förderung der Wissenschaftlichen Forschung | P2ZHP3_195181 | Eric Olinger |

The funders had no role in study design, data collection and interpretation, or the decision to submit the work for publication.

### Author contributions

Jessica J Stanisich, Conceptualization, Data curation, Formal analysis, Validation, Investigation, Visualization, Methodology, Writing - original draft, Writing - review and editing; Dawid S Zyla, Conceptualization, Data curation, Software, Formal analysis, Validation, Investigation, Visualization, Methodology, Writing - original draft, Writing - review and editing; Pavel Afanasyev, Conceptualization, Data curation, Software, Formal analysis, Supervision, Validation, Investigation, Visualization, Methodology, Project administration; Jingwei Xu, Data curation, Validation, Methodology; Anne Kipp, Conceptualization, Data curation, Formal analysis, Validation, Investigation, Visualization; Eric Olinger, Conceptualization, Data curation, Formal analysis, Investigation; Olivier Devuyst, Martin Pilhofer, Conceptualization, Resources, Supervision, Funding acquisition, Writing - original draft, Project administration, Writing - review and editing; Daniel Boehringer, Conceptualization, Resources, Formal analysis, Supervision, Validation, Investigation, Visualization, Methodology, Writing - original draft, Project administration, Writing - review and editing; Rudi Glockshuber, Conceptualization, Resources, Supervision, Funding acquisition, Investigation, Writing - original draft, Project administration, Writing - review and editing

### Author ORCIDs

Jessica J Stanisich (iD) https://orcid.org/0000-0002-2702-8092
Dawid S Zyla (iD) https://orcid.org/0000-0001-8471-469X
Pavel Afanasyev (iD) https://orcid.org/0000-0002-6353-6895
Eric Olinger (iD) http://orcid.org/0000-0003-1178-7980
Olivier Devuyst (iD) http://orcid.org/0000-0003-3744-4767
Martin Pilhofer (iD) http://orcid.org/0000-0002-3649-3340
Rudi Glockshuber (iD) https://orcid.org/0000-0003-3320-3843

### Ethics

Human subjects: The use of human urine samples for UMOD purification was approved by the Ethical Committee of the UCLouvain Medical School in Brussels, Belgium (Project EUNEFRON, 2011/04May/184). All individuals gave written informed consent. The completed "Consent form for publication in eLife" has been filed on the donor's behalf.

### Decision letter and Author response

Decision letter https://doi.org/10.7554/eLife.60265.sa1
Author response https://doi.org/10.7554/eLife.60265.sa2

## Additional files

### Supplementary files

• Supplementary file 1. Stanisich_et_al_2020_supplementary_file_1.pdf. (A) ZP module linker and interface amino acid positions with known variants (gnomAD). (B) Cryo-EM data collection and model refinement statistics. (C) Flowchart for cryoSPARC cryo-EM processing. (D) Flowchart for cisTEM cryo-EM processing.

• Transparent reporting form

### Data availability

Structures and maps presented in this paper have been deposited to the Protein Data Bank (PDB) and Electron Microscopy Data Base (EMDB) with the following accession codes: 3.5 Å cryoSPARC map EMD-11388, the UMOD AU model PDB ID: 6ZS5, elongated model of UMOD filament derived from cryoSPARC map PDB ID: 6ZYA, 4.7 Å cisTEM map EMD-11471.

The following datasets were generated:

| Author(s) | Year | Dataset title | Dataset URL | Database and Identifier |
|---|---|---|---|---|
| Stanisich JJ, Zyla DS, Afanasyev P, Xu J, Pilhofer M, Boehringer D, Glockshuber R | 2020 | 3.5 Å cryo-EM structure of human uromodulin filament core map | http://www.ebi.ac.uk/pdbe/entry/emdb/EMD-11388 | Electron Microscopy Data Bank, EMD-11388 |
| Stanisich JJ, Zyla DS, Afanasyev P, Xu J, Pilhofer M, Boehringer D, Glockshuber R | 2020 | 3.5 Å cryo-EM structure of human uromodulin filament core | http://www.rcsb.org/structure/6ZS5 | RCSB Protein Data Bank, 6ZS5 |
| Stanisich JJ, Zyla DS, Afanasyev P, Xu J, Pilhofer M, Boehringer D, Glockshuber R | 2020 | Extended human uromodulin filament core at 3.5 Å resolution | http://www.rcsb.org/structure/6ZYA | RCSB Protein Data Bank, 6ZYA |
| Stanisich JJ, Zyla DS, Afanasyev P, Xu J, Pilhofer M, Boehringer D, Glockshuber R | 2020 | Extended cryo-EM map of native human uromodulin filament core at 4.7 Å | http://www.ebi.ac.uk/pdbe/entry/emdb/EMD-11471 | Electron Microscopy Data Bank, EMD-11471 |

The following previously published dataset was used:

| Author(s) | Year | Dataset title | Dataset URL | Database and Identifier |
|---|---|---|---|---|
| Karczewski KJ | 2020 | Genome Aggregation Database (GnomAD) | https://gnomad.broadinstitute.org/ | Genome Aggregation Database (GnomAD) v2.1.1, GRCh37 |

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
