## [Decision Letter]

Thank you for submitting your article "The cryo-EM structure of the human uromodulin filament core reveals a unique assembly mechanism" for consideration by *eLife*. Your article has been reviewed by two peer reviewers, including Sjors HW Scheres as the Reviewing Editor and Reviewer #1, and the evaluation has been overseen by John Kuriyan as the Senior Editor. The following individual involved in review of your submission has agreed to reveal their identity: Matthias Wolf (Reviewer #2).

The reviewers have discussed the reviews with one another and the Reviewing Editor has drafted this decision to help you prepare a revised submission.

Summary:

This paper describes cryo-EM reconstructions of the human uromodulin filament core. It follows on from a Science paper by the same group describing lower resolution tomography results and how UMOD filaments constitute a crucial component of the innate immune system against urinary tract infection. The presented structure is novel and unique. Interactions with biological filaments are important disease factors. The authors describe the first structure of the polymerized uromodulin filament core, providing experimental evidence for a β-strand complementing assembly mechanism similar to ones found in chaperone-usher or type V pili. The reviewers agreed that this paper is in principle suitable for *eLife*, but wondered whether the maps could be improved to provide stronger evidence for the proposed atomic model. The reviewers were of the opinion that statements regarding the speculative model should be toned down.

Major points:

1) Both reviewers carefully inspected the submitted cryo-EM reconstructions and models. The maps show pronounced radial blurring with increasing radius, making it difficult to interpret details off-axis. The interpretation of the map in terms of the proposed strand-exchanged model lies close to the limit of what these reviewers are comfortable with. Reviewer #1 was uncomfortable with some of the main-chain tracing decision made (See images 1, 2, 3 and 4); reviewer #2 pointed out that the model is consistent with other strand-exchanged structures, in particular the type V pili, thereby providing further confidence. In a revised version of the manuscript, the authors should do one, or both, of the following two options: 1) improve the quality of the map (see suggestions 2 and 3 below); 2) make additional supplementary figures that support the main-chain tracing in the current model, without hiding the limited quality of the map in the relevant regions.

**Decision letter image 1. sa1fig1:** 

**Decision letter image 2. sa1fig2:** 

**Decision letter image 3. sa1fig3:** 

**Decision letter image 4. sa1fig4:** 

2) Paragraph six: No explanation is provided how the authors arrived at this symmetry parameter. How does the power spectrum look like of aligned averaged filament images? From the small raw micrograph image in Figure 1B it seems clear that the filament may not be strictly flat (rotation = 180 degrees, "snake on a table") but rather slightly twisted as seen in most of the filaments in this image (in particular the one pointing towards the letter B). While the authors attribute the limited quality of the map to the evident flexibility of the filaments, it could also stem from not accounting for the twist mentioned above. How would the maps look like, if the symmetry imposed used a helical parameter near 180 deg (i.e. 63A, 182.5 deg) or (63A, 177.5 deg)? These values are coarsely approximated from Figure 1B, assuming 90 deg twist over 5 scale bars (5*500A/63A = 40 subunits; 90deg/40 = 2.3deg).3) Alternatively, could the maps for specific regions where decisions about the main-chain trace are less clear be further improved by focused refinements on those regions?

4) The model in Figure 4 is based on a linear elongation model first proposed by Shibata et al. for Type V pili. At least, this work should be cited. The former also provided the first experimental validation of the donor strand exchange hypothesis proposed by Waksman et al. (please include citation). As compelling as their model is, the authors provide no evidence (short of detection of tryptic cleavage products by MS in Figure 4—figure supplement 1) for its individual steps. It is so speculative, that they feel compelled to offer an alternative version in the supplement (Figure 4—figure supplement 2A). Short of substantial evidence for this model, this hypothesis should either be greatly simplified and toned down, or left out altogether.

5) The authors use a new density-modification approach in phenix to produce a 4.2A map from the original 4.7A cisTEM map. It is unclear to me how density modification can lead to an increase in resolution. If the authors think this is real, they should explain why this is in a revised version. However, inspection of the map provided for review reveals that most β-strands are not resolved in the density-modified map, suggesting that the resolution is (indeed) still around 4.7A. If so, the 4.7A resolution number should be used.

6) The authors perform 3D classification and analyse heterogeneity in the data by RMSDs among pairs of fitted domains (Figure 1—figure supplement 2). RMSD is a very poor descriptor of the structural heterogeneity. Instead, the authors could calculate angles between subunits, or possibly do a PCA on the multiple structures, to describe the main types of structural variability.

7) Did the authors use Ramachandran restraints in their refinements? If so, this should be stated explicitly, as validation of Ramachandran outliers etc is no longer valid. If Ramachandran statistics are to be used for validation, they should be switched off during model refinement.

---

## [Author Response]

Major points:1) Both reviewers carefully inspected the submitted cryo-EM reconstructions and models. The maps show pronounced radial blurring with increasing radius, making it difficult to interpret details off-axis. The interpretation of the map in terms of the proposed strand-exchanged model lies close to the limit of what these reviewers are comfortable with. Reviewer #1 was uncomfortable with some of the main-chain tracing decision made (See images 1, 2, 3 and 4); reviewer #2 pointed out that the model is consistent with other strand-exchanged structures, in particular the type V pili, thereby providing further confidence. In a revised version of the manuscript, the authors should do one, or both, of the following two options: 1) improve the quality of the map (see suggestions 2 and 3 below); 2) make additional supplementary figures that support the main-chain tracing in the current model, without hiding the limited quality of the map in the relevant regions.

We thank the reviewers for their careful inspection of the models and maps. As was pointed out, the cryo-EM maps indeed contain artifacts, which are caused by the high intrinsic flexibility of the UMOD filament, which leads to a higher resolution at the central axis and lower resolution at the periphery. Due to the flexibility of the filament, helical image reconstruction could not be applied and a number of approaches and software packages were used for solving the structure without applied symmetry (see also point 2). As a result, two best maps with complementary features were obtained: (i) a high resolution cryoSPARC map for the central asymmetric unit resulting from a focused refinement on the filament core. This map displays radial blurring with increased radius, likely due to the relatively small volume included in the focused refinement mask (corresponding to 5 ZP subdomains ~*72*kDa + glycosylation), (ii) A lower-resolution cisTEM map which includes more repeat units with better resolved periphery. In addition, several trials by signal subtraction and focused refinements using the Relion software were performed, however, those did not result in higher resolution maps likely due to the small size of the ZP subdomains (ZPN: 101 residues, ZPC: 128 residues) and the flexibility of subdomain interfaces (see comment 3 for details). The provided maps were of the best quality we could obtain after extensive optimization of masks and refinement parameters. Therefore, we followed the suggestion of the reviewers to support our conclusions by a better illustration of the lower-quality areas of the map (please see Author response images 1, 2 and 3).

Together with images 1, 2 and 3 we would like to make two clarifications:

We created two models (asymmetric unit (AU) and filament model), the latter of which was created by helically expanding the AU along the lower resolution cisTEM map and then refining it again against the 3.5 Å cryoSPARC map, in order to impose refinement restrains at the subunit interfaces. Each of the models has been manually cross-validated. We have added a sentence in the text clarifying that the model and the cryo-EM map should be interpreted on the chains A and D corresponding to the highest resolution of the map (central part) on which the refinement had been focused.

Specifically, to address reviewer 1’s comments about main chain tracing, we would like to reiterate that the AU model (comprised of a ZPN (chain A, residues 328–443), a ZPC (chain D, residues 444–584) and the spanning linker) was built into the center of the high-resolution cryoSPARC map.). The overall folds of these subdomains in the polymerized filament are very similar to the individual ZP subdomains from the previously published crystal structure (PDB: 4WRN; Bokhove et al., 2016, with RMSDs of 2.0 and 1.2 A for ZPN and ZPC, respectively. Additionally, we would like to clarify that the unique subdomain interfaces were built and subsequently refined only for the center of the volume where the map shows the least artifacts (Figure 1—figure supplement 2).

We now provide an additional figure (Figure 1—figure supplement 2) to illustrate the quality of the map in the central core showing the linker region (CS_A_) with surrounding density. We display the Coulomb potential map as a mesh at two different contour levels, 18 σ (red) and 10 σ (blue), to show the continuity of the main chain density.

In Author response image 1 the arrow indicates the region of the map shown in the reviewer’s image; zoomed in views show the higher resolution, symmetry related, part of the map used for structure building.

**Author response image 2. respfig2:** Comparison of the map region affected by radial blurring in the cryoSPARC map with the analogous region in the lower-resolution cisTEM map. (**A**) In image 2 of the attached reviewers’ comments, the map in the loop region 462‒473 is affected by radial blurring in the cryoSPARC map. The main chain trace corresponds well to the lower resolution cisTEM map (**B**), which does not suffer from the same artifacts. (**C**) Additionally, the main chain trace also agrees with the conformation of 462‒743 loop in the ZPC crystal structure (PDB ID: 4WRN; Bokhove et al., 2016, superposition shown in green placed into cisTEM map, as in B). We now omitted the sidechain residues for the loop region 462‒473 from the final model.

Decision letter image 3 – Response: The image provided by the reviewer is a similar view as that Author response image 2; please see our response above.

**Author response image 3. respfig3:** Similar to image 1, image 4 shows the subunit interface at the edge of the reconstructed volume where the map is of poor resolution (indicated with an arrow) compared to the center of the box which we used for building and refinement; however, our AU model was built in the symmetry related part at the center of the 3.5 Å cryoSPARC map (middle panel, highlighted region). We also provide the mesh representation of the cryoSPARC map of the central region shown at two contour levels (red (18 σ), blue (10 σ)) that illustrates a good agreement between the model and the map (right panel).

2) Paragraph six: No explanation is provided how the authors arrived at this symmetry parameter. How does the power spectrum look like of aligned averaged filament images? From the small raw micrograph image in Figure 1B it seems clear that the filament may not be strictly flat (rotation = 180 degrees, "snake on a table") but rather slightly twisted as seen in most of the filaments in this image (in particular the one pointing towards the letter B). While the authors attribute the limited quality of the map to the evident flexibility of the filaments, it could also stem from not accounting for the twist mentioned above. How would the maps look like, if the symmetry imposed used a helical parameter near 180 deg (i.e. 63A, 182.5 deg) or (63A, 177.5 deg)? These values are coarsely approximated from Figure 1B, assuming 90 deg twist over 5 scale bars (5*500A/63A = 40 subunits; 90deg/40 = 2.3deg).

We would like to reiterate that the 2D-classification of the picked UMOD particles revealed highly flexible and bent class-averages of a helical filament formed by repetitive subunits. Consequently, a single-particle analysis with no applied symmetry (C1 symmetry) was used. In order to emphasize this point, we have updated the main text. In initial trials, we attempted to apply helical processing in Relion but these refinements did not yield high resolution reconstructions, likely due to the flexibility of the filament.

Based on the reviewer’s comment we investigated the power spectrum of the 2D classes containing the straightest sections, but due to the small number of monomer repeats (4‒5 per class) it was not possible to interpret the power spectrum in the context of a helical structure, to perform the indexing, and to determine Bessel orders.

Upon further investigation based on the reviewers’ comments, we realized that the previously derived helical parameters had been determined from the high-resolution cryoSPARC map in the real space, using *relion_helix_toolbox*. However, due to the only 2 asymmetric units (an arm, ZPN and neighboring ZPC) in the map we decided to recalculate helical parameters in the same way using the cisTEM map, which includes ~4 asymmetric units. With a search range for helical rise of 60‒67 Å and twist of 175–185°, various cylindrical mask outer diameters (150–250 Å) and percentages of the cropped volume (0.3–0.6) we obtained values of a rise and twist to be ~65 Å and ~180° (with ~1% error), respectively. We updated the values in the manuscript and the Materials and methods section accordingly.

Also, we would like to draw the reviewers’ attention to the subtomogram averages that were previously calculated for direct structural analysis of the UMOD filament by cryo-electron tomography. The subtomogram averages of UMOD filaments show a helical rise of ~6.5 nm and twist of ~180° (Weiss et al., 2020) which is in agreement with the helical parameters obtained from the cisTEM map.

The appearance of different views within some individual filaments, as mentioned by the reviewer, we would attribute to local flexibility (see also answer to point 6).

3) Alternatively, could the maps for specific regions where decisions about the main-chain trace are less clear be further improved by focused refinements on those regions?

As mentioned in the reply to point 1, a focused refinement strategy (with and without signal subtraction; classification with and without alignments and using various masks) had been attempted. Due to high flexibility of the complex, small areas of focus (~12 kDa for individual subdomains of the thin filaments), the signal did not appear to be sufficient to drive refinements and perform better classification in the areas of interest. Taking under consideration our flexibility analysis of UMOD fibers, the resolution in focused refinements with a mask including more than a single subdomain was likely limited by subdomain movement. Also, we would like to point out the challenges related to the specimen preparation, which impose limits on the single-particle analysis of this complex: UMOD appeared to require a relatively thick ice layer to be located on the carbon support as an isolated, intact filament.

4) The model in figure 4 is based on a linear elongation model first proposed by Shibata et al. for Type V pili. At least, this work should be cited. The former also provided the first experimental validation of the donor strand exchange hypothesis proposed by Waksman et al. (please include citation). As compelling as their model is, the authors provide no evidence (short of detection of tryptic cleavage products by MS in Figure 4—figure supplement 1) for its individual steps. It is so speculative, that they feel compelled to offer an alternative version in the supplement (Figure 4—figure supplement 2A). Short of substantial evidence for this model, this hypothesis should either be greatly simplified and toned down, or left out altogether.

We would like to sincerely apologize for the oversight in not including the reference for Shibata et al., 2020. Due to a number of iterations in the manuscript preparation, it had been lost in the originally submitted version of our manuscript. Indeed, we had considered this work in our analysis from the methodological and functional point of view, since it describes a filamentous assembly that is highly analogous to UMOD filaments with respect to β-sheet complementation and proteolysis-dependent assembly competence. We followed the reviewers’ suggestion, curtailed the proposed assembly mechanism in Figure 4 and now emphasize the analogy to the mechanism of type V pilus assembly described by Shibata et al.

5) The authors use a new density-modification approach in phenix to produce a 4.2A map from the original 4.7A cisTEM map. It is unclear to me how density modification can lead to an increase in resolution. If the authors think this is real, they should explain why this is in a revised version. However, inspection of the map provided for review reveals that most β-strands are not resolved in the density-modified map, suggesting that the resolution is (indeed) still around 4.7A. If so, the 4.7A resolution number should be used.

We admit that the density modification approach in Phenix has not yet undergone peer review, and we removed the results of the density-modification in Phenix. Instead, we present the cisTEM map with standard sharpening procedures applied at 4.7 Å resolution. We amended the text and supplementary tables (Figure 1—figure supplement 1, Supplementary file 1B,D) accordingly.

6) The authors perform 3D classification and analyse heterogeneity in the data by RMSDs among pairs of fitted domains (Figure 1—figure supplement 2). RMSD is a very poor descriptor of the structural heterogeneity. Instead, the authors could calculate angles between subunits, or possibly do a PCA on the multiple structures, to describe the main types of structural variability.

We thank the reviewers for this comment and expanded our analysis accordingly. We have utilized the same fitting of individual ZP subdomains as rigid bodies into 3D classes and, instead of using RMSDs, measured the angle between the ZPC subdomains neighboring the central, aligned ZPN subdomain (see updated Figure 1—figure supplement 3). The angular movement of the subdomains of up to 7 degrees, provides a basis to describe filament bending (the angular movement between subdomains propagates along the filament and severely complicates reconstruction of longer filament segments).

Additionally, as suggested, we performed the PCA-based 3D-variablity analysis in the cryoSPARC software. The first component shows a bending motion of the filament similar to our flexibility analysis of 3D classes. We prepared a new figure indicating the main variability components and added a description to the main text (Figure 1—video 2).

7) Did the authors use Ramachandran restraints in their refinements? If so, this should be stated explicitly, as validation of Ramachandran outliers etc is no longer valid. If Ramachandran statistics are to be used for validation, they should be switched off during model refinement.

We have updated the Materials and methods section and the Supplementary file 1B and now explicitly state that the standard settings for refinement in RosettaCM and *phenix.real_space_refine* were used, including Ramachandran restraints.